# Hatchery and Dietary Application of Synbiotics in Broilers: Performance and mRNA Abundance of Ileum Tight Junction Proteins, Nutrient Transporters, and Immune Response Markers

**DOI:** 10.3390/ani14060970

**Published:** 2024-03-20

**Authors:** Mallory B. White, Ali Calik, Rami A. Dalloul

**Affiliations:** 1School of Science, Technology, Engineering & Mathematics (STEM), Virginia Western Community College, Roanoke, VA 24015, USA; mwhite@virginiawestern.edu; 2Avian Immunobiology Laboratory, Department of Poultry Science, University of Georgia, Athens, GA 30602, USA; ali.calik@uga.edu; 3Department of Animal Nutrition & Nutritional Diseases, Faculty of Veterinary Medicine, Ankara University, Ankara 06110, Turkey

**Keywords:** synbiotic, hatchery, broiler, immune response, nutrient transporter

## Abstract

**Simple Summary:**

A single post-hatch oral inoculation of synbiotics, along with its continued use in diets, could be a valuable technique for enhancing industry efficiency and overall outcomes. This method could potentially lead to the earlier colonization of the gastrointestinal tract (GIT) with beneficial microbes and promote a more favorable gut microflora throughout the production cycle. Our results showed that hatchery and dietary synbiotic application have a potential beneficial impact on broiler intestinal immunity by regulating the innate immune response and improving feed conversion. Future research involving extended grow-out studies with a disease challenge would expand on the implications of the early application of synbiotics and provide additional insight into their potential field applications.

**Abstract:**

This study investigated the effects of a synbiotic consisting of inulin, *Enterococcus faecium*, *Pediococcus acidilactici*, *Bifidobacterium animalis*, and *Lactobacillus reuteri* given orally to day (d)-of-hatch (DOH) broiler chicks at the hatchery and in the feed for a 21 d period. A total of 480 Cobb male broilers were randomly divided into one of four treatments using a 2 × 2 factorial design as follows: (1) control (CTRL) group receiving a gel-only oral application on DOH at the hatchery prior to transport and a non-medicated basal corn/soybean meal starter diet; (2) hatchery synbiotic (HS) receiving an oral gel containing the synbiotic (0.5 mL/bird) at the hatchery and the basal diet; (3) CTRL + dietary synbiotic at 0.5 kg/MT (DS); and (4) HS + dietary synbiotic at 0.5 kg/MT (HSDS). On d 7 and d 21, one bird per pen (eight replicate pens/group) was euthanized, and the ileum was immediately removed for qPCR analysis. Data were subjected to a 2-way ANOVA using GLM procedure (JMP Pro17). A significant diet × hatchery interaction was observed in feed conversion ratio (FCR) from d 14 to d 21 (*p* = 0.013) where the HS, DS, and HSDS treatments had a significantly lower FCR compared to the CTRL. However, no significant interaction effect was observed for body weight gain (BWG) or FCR during the overall experimental period. No significant interaction was observed in mRNA abundance of the evaluated genes in the ileum on d 7 and d 21. Gel application with the synbiotic significantly reduced sodium-dependent glucose cotransporter 1 (SGLT1) mRNA abundance on d 7 (*p* = 0.035) in comparison to birds receiving gel alone. Regardless of hatchery application, dietary synbiotic supplementation significantly reduced Toll-like receptor (TLR)2, TLR4, and interleukin (IL)-10 mRNA abundance on d 7 (*p* = 0.013). In conclusion, these findings showed that hatchery and dietary synbiotic application could have a potential beneficial impact on broiler intestinal immunity by regulating the TLR response, a key element of innate immunity. FCR was improved from d 14 to d 21 after synbiotic application. Future research involving extended grow-out studies with a disease challenge would expand on the implications of an early application of synbiotics.

## 1. Introduction

Reduced antibiotic use in food animal production does not eliminate the need to curtail disease and promote animal health [1]. Although antibiotic resistance in bacteria can often be attributed to improper human use [2], increased public pressure on agricultural use has led to more regulation and voluntary reduction of antibiotic use by the food animal industry [3]. Ultimately, the decline in using antibiotics has elevated the interest in alternatives including probiotics, prebiotics, and synbiotics to achieve a healthy gut for better performance, optimal nutrient utilization, and disease resistance.

In the poultry industry, it is standard practice to transport chicks from hatcheries to commercial houses without feed or water for up to 48 h post-hatch [4]. Although chicks have yolk sac reserves to sustain them, negative consequences including reduced body weight (BW) at placement, decreased post-hatch growth, higher mortality, and delayed intestinal maturation have been documented in chicks that experience any delay to feed or water for a prolonged period of time [5,6,7,8]. Moreover, in intensive broiler production systems, newly hatched birds initially encounter bacteria in the hatchery and the house environment, as opposed to that of the hen or nest material. Consequently, this results in the inadequate bacterial colonization of the chicks’ gastrointestinal tracts (GITs), rendering them more susceptible to intestinal disorders [9]. Early intestinal colonization with beneficial bacteria not only helps prevent pathogenic bacteria-related intestinal disorders, but also improves intestinal maturation and integrity [10,11]. Hence, the concept of early nutrition, including in ovo nutrition and early feed access, becomes an important strategy to achieve industrial goals. This concept is dependent on providing nutrients and/or active compounds either during late embryogenesis or immediately after hatching [12,13]. Therefore, a careful and timely manipulation of the embryo or newly hatched chicks with such compounds could help to enhance their strength and adaptability, which, in turn, would support their future performance.

Probiotics, prebiotics, and synbiotics are important nutraceuticals used in poultry diets to stimulate gastrointestinal development and immunological activity in birds by conferring a healthy enteric microbial balance [14,15,16]. While a prebiotic is defined as a substance that supports the growth of probiotics, the beneficial live microorganisms, a synbiotic is a substance that contains both prebiotics and probiotics. In recent years, in ovo and early post-hatch use of these compounds has been investigated to determine their potential effects during both early life and the grow-out period. Accordingly, the early introduction of these compounds can improve intestinal microstructure [11,17], alter immune response [18,19], and modify intestinal microbiota with more favorable bacteria [20,21].

The importance of the early introduction of probiotics, prebiotics, and synbiotics for broiler performance and health has been well-documented and their positive impact on intestinal health long recognized. However, in comparison to in ovo inoculation, less information is available regarding the hatchery application of such compounds before transport and their continued dietary use. Based on previous findings suggesting the positive influence of the early beneficial bacterial colonization of the GIT on broiler health and performance, the current study hypothesized that hatchery application and the subsequent dietary use of a well-balanced synbiotic consisting of inulin, *Enterococcus faecium*, *Pediococcus acidilactici*, *Bifidobacterium animalis*, and *Lactobacillus reuteri* may be an effective method to improve broiler performance by modifying intestinal integrity, immunity, and function.

## 2. Materials and Methods

### 2.1. Birds, Management, and Experimental Diets

This project was approved and conducted under the guidelines of Institutional Animal Care and Use Committee (IACUC #18-037). In total, 480 d old Cobb male broilers were obtained from a commercial hatchery, randomly divided in half, and received either an individual oral dose of 0.5 mL gel without the synbiotic (gel only) or 0.5 mL gel containing soluble synbiotic (16 g soluble synbiotic/1 L gel). Each dose was given using a syringe and an oral gavage needle. Prior to pick up, chicks were vaccinated in ovo against Marek’s disease virus at the hatchery following standard commercial practices. Chicks were then transported for 6 h in hatch trays to mimic industry transportation methods. Upon arrival at the animal research facility, each group of birds was randomly divided in half and assigned to either basal dietary treatment or a synbiotic-supplemented basal diet (Table 1) to yield the four treatment groups. This 2 × 2 factorial design consisted of diet (basal with or without synbiotic) and hatchery application (oral gel application with or without synbiotic) for a total of 4 treatments (8 replicate pens/treatment; 15 birds/replicate; 120 birds/treatment): (1) control (CTRL) group receiving a gel-only oral application on DOH at the hatchery prior to transport and a non-medicated basal corn/soybean meal starter diet; (2) hatchery synbiotic (HS) receiving an oral gel containing the synbiotic (0.5 mL/bird) at the hatchery and the basal diet; (3) CTRL + dietary synbiotic at 0.5 kg/MT (DS); and (4) HS + dietary synbiotic at 0.5 kg/MT (HSDS). The synbiotic used in the current study consisted of dehydrated probiotic bacterial species (minimum 10^8^ CFU/g) including *Enterococcus faecium*, *Pediococcus acidilactici*, *Bifidobacterium animalis*, *Lactobacillus reuteri*, and the prebiotic inulin (fructooligosaccharides).

Birds were housed in floor pens with fresh pine shavings, adjustable nipple drinkers, and plastic bucket-type feeders. Water and feed were provided ad libitum throughout the experimental period. The temperature began at 32 °C and gradually adjusted down until reaching 24 °C at d 21. Light was provided for 24 h during the first week and gradually adjusted down from d 7 through d 14 to 4 h of dark and 20 h of light where it remained until the end of the experiment. Broiler chickens were weighed on a per-pen basis, and feed intake (FI) was recorded at weekly intervals. Any mortality was removed and recorded (including bird weight) twice daily. Mortality-corrected body weight gain (BWG), FI, and feed conversion ratio (FCR) were subsequently calculated to evaluate growth performance.

### 2.2. Total RNA Extraction and Reverse Transcription

On d 7 and d 21, one bird per pen was euthanized; the mid-ileum (~10 cm section) was immediately removed, rinsed with 10% ice-cold PBS, snap-frozen in liquid nitrogen, and stored at −80 °C until analysis. Tissue samples were homogenized using a TissueLyser (QIAGEN, Valencia, CA, USA), and RNA was extracted following standard manufacturer’s protocol (Qiagen RNeasy Mini Kit, Hilden, Germany). Optical density (OD) was used to assess RNA quality (OD 260/280) and concentration (OD 260) as measured by a NanoDrop spectrophotometer (Thermo Fisher Scientific, Waltham, MA, USA). After extraction, 2 µg of total RNA was used to synthesize first-strand cDNA using High Capacity cDNA Reverse Transcription kit (Applied Biosystems, Foster City, CA, USA) according to the manufacturer’s recommendation, and the cDNA was stored at −20 °C.

### 2.3. Quantitative Real-Time PCR

The mRNA abundance of tight junction (TJ) proteins [zona occludens (ZO)-1, ZO-2, and claudin (CLDN)-1], nutrient transporters [peptide transporter 1 (PepT1), sodium-dependent glucose cotransporter 1 (SGLT1)], and immune response markers interleukin (IL)-10, Toll-like receptor (TLR)2 and TLR4 were determined by quantitative real-time PCR (ABI 7500 Fast Real-Time PCR System, Applied Biosystems) using Fast SYBR™ Green Master Mix (Applied Biosystems). Primer details are provided in Table 2. Product specificity was confirmed by the analysis of the melting curves produced by ABI 7500 software (version 2.0.3). The cDNA was diluted to 1:20 in nuclease-free water, and 2 μL of the diluted cDNA were added to each well of a 96-well plate. Next, 8 μL of real-time PCR master mix containing 5 μL of Fast SYBR Green Master Mix, 0.5 μL each of 2 μM forward and reverse primers, and 2 μL of sterile nuclease-free water per reaction were added to each well for a final volume of 10 μL. Changes in the mRNA abundance of each target gene were calculated using glyceraldehye-3-phosphate dehydrogenase (GAPDH) as the endogenous control. Results for each treatment are reported as average mRNA abundance relative to GAPDH using the 2^−ΔΔCt^ method [22]. The calibrator for each gene was the average ΔCt value from the CTRL for each sampling day.

### 2.4. Statistical Analysis

Data analysis was performed using two-way ANOVA (JMP Pro17) between diet and hatchery applications using Fisher’s least significant difference (LSD) method. The results report the interaction between diet and application, the main effect of diet, and the main effect of hatchery application. The results are reported as least square means (LS Means) with standard error means (SEM). The probability *p* < 0.05 was considered significant unless otherwise noted.

## 3. Results

### 3.1. Growth Performance

The effects of hatchery synbiotic application and subsequent post-hatch dietary synbiotic supplementation on the growth performance of chickens are shown in Table 3. There was no significant diet × hatchery interaction effect on performance parameters including BWG, FI, and FCR during d 0–7, d 7–14, and d 0–14 of the study. There was a difference due solely to the main effect of hatchery application during the first week where the BWG of the group provided the gel with synbiotic was significantly lower than the group provided the gel alone (*p* = 0.020). Moreover, a main effect of diet (*p* = 0.019) was observed where birds provided the basal diet had a higher FI than the supplemented birds during d 0–7. Neither hatchery nor diet application influenced FCR during the first two weeks of the study.

Similarly, no significant diet × hatchery interaction effect was observed in BWG among the treatment groups during d 14–21 and for the overall period (d 0–21). However, there was a significant difference due to the interaction of diet × hatchery application from d 14 to 21 (*p* = 0.003) and overall period (*p* = 0.014) in which CTRL had a higher FI compared to HS and DS, yet HSDS had a similar FI compared to all other treatments. Due to the reduced FI and similar BWG, a significant diet × hatchery interaction was observed in FCR from d 14 to 21 (*p* = 0.013) where the HS, DS, and HSDS treatments had a significantly lower FCR compared to the CTRL. However, this interaction was not apparent during the overall experimental period. When looking at the main effect of hatchery application from d 14 to d 21, birds given gel with a synbiotic had a lower FCR compared to birds given gel without a synbiotic (*p* = 0.042). There were no main effects of hatchery or diet on performance parameters during the overall study.

### 3.2. mRNA Abundance of Gut Integrity, Nutrient Transporters, and Immune Response-Related Genes in the Ileum on d 7 and d 21

The effects of hatchery synbiotic application and subsequent post-hatch dietary synbiotic supplementation on mRNA abundance of gut integrity, nutrient transporters, and immune response-related genes in the ileum of broilers are shown in Figure 1, Figure 2, and Figure 3, respectively. No significant interaction was observed in the mRNA abundance of the evaluated genes in the ileum on d 7 and d 21. Gel application with a synbiotic significantly downregulated SGLT1 mRNA abundance on d 7, but not on d 21 where there was a trend for synbiotic supplemented groups to have a numerically higher expression than CTRL (*p* = 0.082). Regardless of hatchery application, dietary synbiotic supplementation significantly downregulated the mRNA abundance of IL-10 (*p* = 0.040), TLR2 (*p* = 0.037), and TLR4 (*p* = 0.013) on d 7.

## 4. Discussion

Modern commercial broilers are routinely exposed to a myriad of environmental stressors including handling and transport as well as delayed access to feed and water. The time baby chicks spend from the hatchery to the grow-out facility, including transportation, can be up to 48 h, and chicks that hatch early spend additional time in the hatchery without feed and water access [6]. A recent study reported that any delay to feed and water over 24 h significantly reduces the d 35 final BW in broilers [6]. Therefore, early nutritional strategies can help in alleviating the negative effects of delayed access to external nutrients and the post-hatch environmental stressors [12]. As commercial broilers reach market age within a short period of time, it may be advantageous to have the application of probiotics, prebiotics, and synbiotics at the hatchery to establish a beneficial gut microbiota early in production. It is known that intestinal microbiota composition and their metabolites play a pivotal role in the nutrient uptake, intestinal integrity, and immune function of broilers. In this regard, the application of beneficial bacteria before or right after hatch could help to establish a balanced microbiota to support gut health and broiler performance [23]. In this context, this present study aimed to determine the effects of a single post-hatch oral inoculation of synbiotics before a 6 h transportation time, and subsequent dietary synbiotic supplementation, on broiler performance and the mRNA abundance of immune response, tight junction proteins, and nutrient transporter-related genes in the ileum.

Contrary to our hypothesis, there was no significant change in final BWG and FCR due to the interactions of diet and hatchery application. Moreover, the interactive effect of hatchery and dietary synbiotic application on FCR was only prominent during d 14–21 in the HS, DS, HSDS groups compared to CTRL. While published studies have mostly focused on in ovo nutrition, there is very little information about the hatchery application of synbiotics as an early nutritional strategy. According to a previous study, in ovo application and the continued dietary use of a synbiotic (*E. faecium* and a prebiotic inulin) were reported to be insufficient to promote overall (d 0–42) broiler performance [11]. Similarly, in ovo inoculation of different synbiotics on embryonic day 12 (E12) did not influence broiler performance post hatch [24,25,26]. However, the early introduction of these compounds was reported to have more pronounced effects on broiler performance under challenge conditions [27,28]. Even though synbiotic treatment did not influence d 21 BW, it seemed to lower both BW and BWG for the first two weeks. The observed reduction in early performance might be related to microbial growth in the intestine of synbiotic groups, which could divert energy resources away from tissue and muscle development. This finding is also supported by other studies where conventional birds exhibited a slower growth rate compared to germ-free birds during the first few weeks [10,29]. However, birds compensated for the reduced BW during the third week and exhibited better feed efficiency. The absence of a growth-promoting effect of synbiotic treatment (hatchery and/or dietary) is possibly attributable to the study’s design which lacked any major stress or challenge conditions. Such feed additives tend to be more effective under field conditions [30] or disease challenge models [28,31,32]. Overall, the results of the current study provide preliminary insights into the potential impact of synbiotics in hatchery applications and their role in broiler growth under normal conditions. Further research is warranted to investigate their effects on broiler performance through market age, particularly in response to disease challenges.

TJ proteins are paramount in ensuring the structural integrity and selective permeability of epithelial barrier, playing a pivotal role in maintaining tissue homeostasis [33]. Their biological activities extend to crucial functions such as preventing the invasion of pathogens, regulating nutrient absorption, and influencing various cellular processes, making them indispensable for overall tissue function and health [31]. Early beneficial bacterial colonization in the intestine can influence TJ structure and/or function, thereby promoting intestinal maturity and gut barrier function [8,34]. Neither hatchery application nor the continued dietary use of synbiotics made a valuable contribution to the mRNA abundance of ileal TJ proteins in the absence of a challenge during this study. However, the dietary use of synbiotics influenced the mRNA abundance of intestinal TJ proteins under disease challenge [35] or heat stress conditions [36]. Therefore, the observed findings might be associated with the controlled experimental conditions, which provided a less stressful environment with fewer challenges from potential pathogens.

Intestinal immunity is crucial, serving as the frontline defense against infections, regulating inflammation, and fostering a healthy balance with resident microbiota [37]. Considering the close relationship between the intestinal surface and the diverse microbiota, it is not surprising that innate signals play a crucial role in maintaining intestinal balance [38]. TLRs, a group of innate molecules, recognize conserved molecular patterns expressed on a broad spectrum of microbial fragments, including antigens from the microbiota and potential invading pathogens [39]. Regardless of hatchery application, dietary synbiotic supplementation significantly reduced the mRNA abundance of TLR2, TLR4, and IL-10 on d 7. The expression of TLRs is generally related to increased pathogens and antigenic load in the intestine [40], microbial dysbiosis [41], and presence of intestinal stressors [42,43]. Additionally, activated TLRs elicit an increased expression of pro-inflammatory factors that may compromise gut health [44]. Similar results were reported by Pender et al. [45] where birds received probiotics through in ovo application at different inclusion rates. Therefore, the findings of the current study may be attributed to a more balanced intestinal microbiota, as dietary synbiotics are reported to reduce several intestinal pathogens [46]. However, a more meaningful comparison of these results could be achieved through microbiome analysis, providing a clearer understanding of their functional impact.

Nutrient uptake in the intestine is mainly mediated by several transport proteins [47]. These transporters are highly responsive to various external factors including intestinal microbiota, nutrient availability, and stress [47,48,49]. SGLT1 facilitates the transport of glucose and galactose across the apical side of intestinal epithelial cells [50]. Birds administered the gel containing a synbiotic showed a lower SGLT1 mRNA abundance on d 7 that correlated with the reduced BWG during the first week. The observed decrease could be attributed to developing bacterial populations in the synbiotic groups utilizing a portion of dietary glucose during the early stages, leading to less glucose being available for luminal absorption. Conversely, all synbiotic-treated groups tended to exhibit higher SGLT1 mRNA abundance compared to the control by the end of week three (*p* = 0.082). This finding also correlated with the improved FCR and compensated for BWG during d 14–21. The disparity in outcomes between d 7 and d 21 might have stemmed from the presence of established beneficial bacteria in the mucosal layer, as the gut microbiota in poultry typically matures between days 14 and 21 post-hatch [51].

## 5. Conclusions

The perinatal period spanning from late-term embryo to few days post-hatch is an important period for the development of the GIT and the immune system of poultry. Therefore, establishing beneficial microbiota in the chicken intestine early in production is one of the industry’s goals to contribute to broilers’ overall health and performance. Despite limited evidence regarding hatchery synbiotic application, this preliminary study obtained significant results by influencing growth performance and immunity. Nevertheless, additional research is warranted to explore the efficacy of this application method under clinical or sub-clinical disease models, along with a detailed microbiome analysis in broilers raised to market age.

## Figures and Tables

**Figure 1 animals-14-00970-f001:**
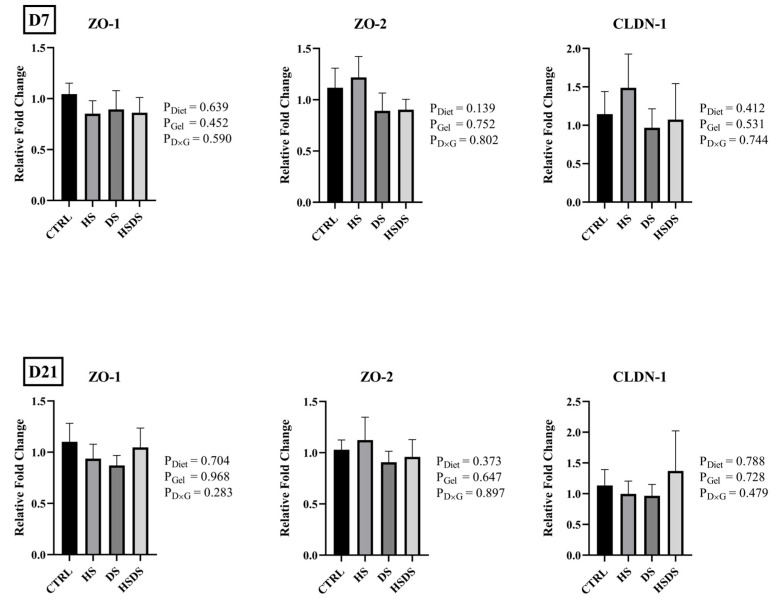
Effects of hatchery synbiotic administration and post-hatch dietary synbiotic supplementation on the relative mRNA abundance of tight junction protein-related genes in the ileum on d 7 and 21. Main effects of diet, hatchery applications, and the interactions are presented next to each graph. Data are presented as mean ± SEM (*n* = 8 birds/group).

**Figure 2 animals-14-00970-f002:**
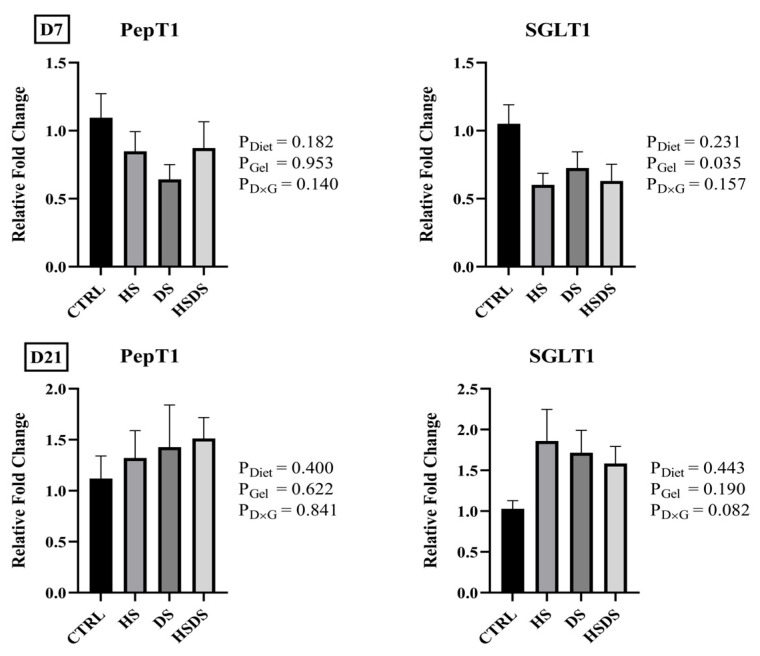
Effects of hatchery synbiotic administration and post-hatch dietary synbiotic supplementation on relative mRNA abundance of nutrient transporter-related genes in ileum on d 7 and 21. Main effects of diet, hatchery applications, and the interactions are presented next to each graph. Data are presented as mean ± SEM (*n* = 8 birds/group).

**Figure 3 animals-14-00970-f003:**
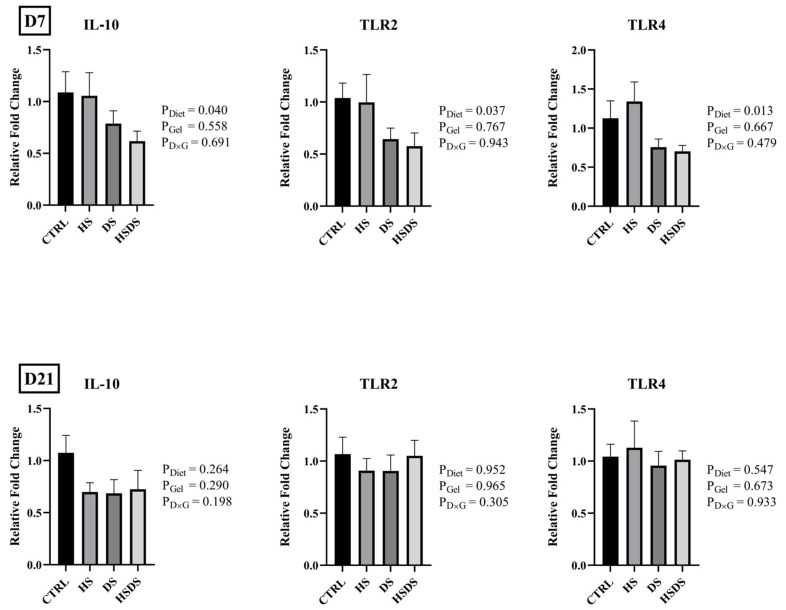
Effects of hatchery synbiotic administration and post-hatch dietary synbiotic supplementation on relative mRNA abundance of immune response-related genes in ileum on d 7 and 21. Main effects of diet, hatchery applications, and the interactions are presented next to each graph. Data are presented as mean ± SEM (*n* = 8 birds/group).

**Table 1 animals-14-00970-t001:** Feed composition of broiler starter diet (d 0–21).

Ingredient, %	0–21 d
Corn	59.73
Soybean meal, CP 48%	33.20
Vegetable oil	3.00
Limestone	0.68
Dicalcium phosphate	2.20
DL-Methionine	0.20
L-Lysine HCl	0.24
L-Threonine	0.09
Salt	0.30
Vitamin–Mineral Premix (NB3000) ^1^	0.36
Total	100.0
Chemical composition (Calculated)	
Dry Matter, %	87.75
Crude Protein, %	21.00
AME_n_, kcal/kg	3060
Lysine, %	1.32
Dig. Lysine, %	1.18
Methionine + cysteine, %	0.88
Dig. Methionine + cysteine, %	0.78
Threonine, %	0.90
Dig. Threonine, %	0.77
Calcium, %	0.90
Available phosphorus, %	0.45

^1^ Vitamins per kg diet: retinol 3.33 mg, cholecalciferol 0.1 mg, α-tocopherol acetate 23.4 mg, vitamin K3 1.2 mg, vitamin B1 1.6 mg, vitamin B2 9.5 mg, niacin 40 mg, pantothenic acid 9.5 mg, vitamin B6 2 mg, folic acid 1 mg, vitamin B12 0.016 mg, biotin 0.05 mg, choline 556 mg. Minerals per kg diet: Mn 144 mg, Fe 72 mg, Zn 144 mg, Cu 16.2 mg, I 2.1 mg, Se 0.22 mg.

**Table 2 animals-14-00970-t002:** Sequences of primer pairs used for amplification of target and reference genes. ^1^

Gene ^2^	Primer Sequence (5′–3′)	Size (bp)	Acc (Reference)
GAPDH	CCTAGGATACACAGAGGACCAGGTT	64	NM_204305
	GGTGGAGGAATGGCTGTCA		
ZO-1	GGAGTACGAGCAGTCAACATAC	101	XM_413773
	GAGGCGCACGATCTTCATAA		
ZO-2	GCGTCCCATCCTGAGAAATAC	89	NM_205149.1
	CTTGTTCACTCCCTTCCTCTTC		
CLDN-1	GTGTTCAGAGGCATCAGGTATC	107	NM_001013611.2
	GTCAGGTCAAACAGAGGTACAA		
PepT1	CCCCTGAGGAGGATCACTGTT	66	NM_204365
	CAAAAGAGCAGCAGCAACGA		
SGLT1	GCCATGGCCAGGGCTTA	71	XM_415247
	CAATAACCTGATCTGTGCACCAGTA		
TLR2	GCGAGCCCCCACGAA	61	NM_204278
	GGAGTCGTTCTCACTGTAGGAGACA		
TLR4	CCACACACCTGCCTACATGAA	63	NM_001030693
	GGATGGCAAGAGGACATATCAAA		
IL-10	CGCTGTCACCGCTTCTTCA	63	NM_001004414
	CGTCTCCTTGATCTGCTTGATG		

^1^ For each gene, the primer sequence for forward (F) and reverse (R) (5′-3′) primers, the amplicon size (bp), and the NCBI accession number (Acc) used for the primer design are listed. ^2^ GAPDH: Glyceraldehyde 3-phosphate dehydrogenase; ZO: Zonula occludens; CLDN: Claudin; PepT1: Peptide Transporter 1 (SLC15A1); SGLT1: Sodium-dependent glucose cotransporter 1 (SLC5A1); TLR: Toll-like receptor; IL: Interleukin.

**Table 3 animals-14-00970-t003:** Effects of hatchery synbiotic administration and post-hatch dietary synbiotic supplementation on broiler performance. ^1^

	Treatments ^2^				Diet ^3^		Hatchery ^3^			Statistics		
	Basal Diet		Synbiotic Diet							*p*-Value		
Item	Gel- (CTRL)	Gel+ (HS)	Gel- (DS)	Gel+ (HSDS)	Basal	Synbiotic	Gel-	Gel+	RMSE ^4^	Diet	Gel	D × G
0 to 7 d												
BW d 0 (g)	43.11	42.70	42.81	42.64	42.91	42.73	42.96	42.67	0.72	0.486	0.262	0.631
BWG (g)	118.9	110.5	110.1	107.7	114.7	109.3	114.9	109.1	6.55	0.031	0.020	0.286
FI (g)	161.5	160.4	154.1	151.8	160.9	153.0	157.8	156.0	8.35	0.019	0.609	0.857
FCR	1.36	1.43	1.39	1.41	1.39	1.40	1.37	1.42	0.08	0.557	0.058	0.333
7 to 14 d												
BWG (g)	307.8	276.9	287.5	288.1	292.4	287.8	297.7	282.5	24.58	0.608	0.097	0.087
FI (g)	399.8	369.7	376.5	376.3	384.8	376.4	388.1	373.0	22.03	0.300	0.067	0.070
FCR	1.30	1.34	1.31	1.31	1.32	1.31	1.31	1.33	0.08	0.837	0.488	0.530
0 to 14 d												
BWG (g)	426.7	387.5	398.5	395.8	407.1	397.2	412.6	391.6	29.81	0.362	0.061	0.100
FI (g)	561.2	538.3	530.6	528.2	549.8	529.4	545.9	533.2	28.82	0.059	0.233	0.331
FCR	1.32	1.39	1.33	1.34	1.36	1.34	1.33	1.37	0.08	0.529	0.144	0.218
14 to 21 d												
BWG (g)	505.9	484.7	487.2	504.0	495.3	495.6	496.5	494.3	31.49	0.980	0.849	0.105
FI (g)	732.3 ^a^	647.2 ^b^	664.0 ^b^	694.9 ^ab^	689.8	679.4	698.1	671.0	49.53	0.567	0.140	0.003
FCR	1.45 ^a^	1.34 ^b^	1.37 ^b^	1.38^b^	1.39	1.37	1.41	1.36	0.06	0.406	0.042	0.013
0 to 21 d												
BWG (g)	932.6	872.2	885.7	899.8	902.4	892.7	909.2	886.0	58.44	0.650	0.280	0.087
FI (g)	1293.6 ^a^	1185.5 ^b^	1194.5 ^b^	1205.5 ^ab^	1239.5	1208.8	1244.0	1204.3	72.08	0.246	0.137	0.014
FCR	1.39	1.36	1.35	1.36	1.37	1.36	1.37	1.36	0.05	0.344	0.654	0.350
Mortality, %	4.17	1.67	2.50	1.67	2.92	2.08	3.33	1.67	-			

^1^ CTRL: basal diet and gel application only; HS: hatchery synbiotic; DS: dietary synbiotic; HSDS: hatchery synbiotic and dietary synbiotic. Gel-: gel without synbiotic; Gel+: gel containing synbiotic. Data presented as least square means (LSM). Differing letters in the rows represent treatments that are significantly different from one another. ^2^ Data represent mean values of 8 replicates per treatment. ^3^ Data represent mean values of 16 replicates per treatment. ^4^ RMSE: Root Mean Square Error.

## Data Availability

None of the data were deposited in an official repository.

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
