# Peer review of "Hatchery and Dietary Application of Synbiotics in Broilers: Performance and mRNA Abundance of Ileum Tight Junction Proteins, Nutrient Transporters, and Immune Response Markers"

_animals, 2024, doi:10.3390/ani14060970_

Round 1
Reviewer 1 Report
Comments and Suggestions for Authors
The work is interesting and brings new elements regarding the administration of synbiotics to chickens directly in the hatchery in order to early colonize the intestinal microflora. On the basis of well-selected literature, the authors justified the purposefulness of the conducted research. The aim of the work was clearly formulated. The material used for the research is sufficient, the research methods have been selected appropriately and described in detail. The results were presented in 3 tables and 3 figures and discussed in detail.
Why did the authors finish the study on day 21 of broiler chicken rearing and not on day 35 or 42? Perhaps the effect of the administered synbiotics would be more significant.
The conclusions are correct and result from the obtained research results.
Reviewer 2 Report
Comments and Suggestions for Authors
This purpose of this study was to evaluate the effect of a synbiotic administered orally at hatch followed by a 6h transport +/- dietary inclusion of the same synbiotic for 21-d post hatch on performance and ileal gene expression in broiler chickens.
Title: Suggest modifying the title to be more specific to the study - Hatchery and dietary application of a synbiotic
Simple summary:
Line 17: Please remove guarantee. It would be more appropriate to use suggest or a similar word.
Line 19: Summarize the impact of treatment on performance.
Abstract:
A brief description of the synbiotic is needed. What is the composition of it? Is it commercially available?
Line 39: Confirm p-value please.
Line 41: A summary regarding performance would be good to include.
Introduction:
Lines 72-77: If possible, please include more information about hatchery application of pre-, pro, and/or synbiotics. More specific information about dietary inclusion would be useful as well. Clearly define synbiotics here since this is the purpose of your study. As currently written, pre-, pro-, and synbiotics are vaguely discussed as being quite similar. It could mislead readers that lack knowledge on the subject.
Line 86: Please clarify what you mean by "well-balanced". Again, this is vague. A specific description of the synbiotic used in your study is needed.
Materials and methods:
Line 90: Include approval #.
Line 91: Did the chicks receive any vaccinations, etc.? Please clarify.
Line 92+: What was the composition of the gel? Is it commercially available? Is the synbiotic commercially available? How did you select the dose? Why did you select this synbiotic? How much inulin was included in the mixture or is this proprietary? How did you prepare the mixture? On site at the hatchery? I know the gel was administered orally. I assume by oral gavage using a gavage needle. Please specify. How did you select the synbiotic concentration for the diet? Did you evaluate gut colonization at hatch to see what the chicks were colonized with at hatch prior to synbiotic application? This information is an essential component of your materials and methods.
Line 125: Approximate length of ileum removed?
Results:
Table 3. Did you look at standard error for each group? Quite a large reduction in performance between the gel + / basal diet compared to basal diet only.
Discussion:
The discussion is well written. Please specify what E12 is in Line 255. I know but others may not.
Reviewer 3 Report
Comments and Suggestions for Authors
This study suggests that hatchery and dietary addition of synthetic bacteria may have potential beneficial effects on intestinal immunity and growth performance of broilers by regulating TLR response. The logic of the whole study is clear and clear. Although the amount of data is small, it can fully explain the biological problems. It can provide theoretical reference for improving intestinal health, nutrient utilization and disease resistance of broilers in practice. However, there are some doubts that need to be explained.
1. Why did this study choose the 7th and 21st day as two time nodes for sampling and detection of key intestinal genes?
2. Differences in the expression of key intestinal proteins were detected in the study, and their key role in the structural integrity and selective permeability of the epithelial barrier was also mentioned in the discussion by the authors. However, the addition of intestinal tissue-based section staining will provide a more intuitive monitoring of intestinal structure and appearance changes.
3. Chicken resistance to environmental pathogens and growth cycles are heavily influenced by chicken breed, and differences between traditional commercial broilers and high-quality local broilers can be significant. Therefore, the chicken breeds used in this experiment can be properly introduced.
4. The bar chart in the study suggests adjustments. It is not necessary to fully display P-values for data with insignificant differences. Identifying significant differences by "letters" or "*" helps readers understand quickly.
